# Densely Connected Attention Propagation for Reading Comprehension

*Yi Tay[1], *Luu Anh Tuan[2], Siu Cheung Hui[3] and Jian Su[4]

[1,3]Nanyang Technological University, Singapore
[2,4]Institute for Infocomm Research, Singapore

ytay017@e.ntu.edu.sg[1]
at.luu@i2r.a-star.edu.sg[2]
asschui@ntu.edu.sg[3]
sujian@i2r.a-star.edu.sg[4]

## Abstract

We propose DECAPROP (*Densely Connected Attention Propagation*), a new densely connected neural architecture for reading comprehension (RC). There are two distinct characteristics of our model. Firstly, our model densely connects all pairwise layers of the network, modeling relationships between passage and query across all hierarchical levels. Secondly, the dense connectors in our network are learned via attention instead of standard residual skip-connectors. To this end, we propose novel *Bidirectional Attention Connectors* (BAC) for efficiently forging connections throughout the network. We conduct extensive experiments on four challenging RC benchmarks. Our proposed approach achieves state-of-the-art results on all four, outperforming existing baselines by up to $2.6\% - 14.2\%$ in absolute F1 score.

## 1 Introduction

The dominant neural architectures for reading comprehension (RC) typically follow a standard 'encode-interact-point' design [Wang and Jiang, 2016; Seo et al., 2016; Xiong et al., 2016; Wang et al., 2017c; Kundu and Ng, 2018]. Following the embedding layer, a compositional encoder typically encodes $Q$ (query) and $P$ (passage) individually. Subsequently, an (bidirectional) attention layer is then used to model interactions between $P/Q$. Finally, these attended representations are then reasoned over to find (point to) the best answer span. While, there might be slight variants of this architecture, this overall architectural design remains consistent across many RC models.

Intuitively, the design of RC models often possess some depth, i.e., every stage of the network easily comprises several layers. For example, the R-NET [Wang et al., 2017c] architecture adopts three BiRNN layers as the encoder and two additional BiRNN layers at the interaction layer. BiDAF [Seo et al., 2016] uses two BiLSTM layers at the pointer layer, etc. As such, RC models are often relatively deep, at the very least within the context of NLP.

Unfortunately, the depth of a model is not without implications. It is well-established fact that increasing the depth may impair gradient flow and feature propagation, making networks harder to train [He et al., 2016; Srivastava et al., 2015; Huang et al., 2017]. This problem is prevalent in computer vision, where mitigation strategies that rely on shortcut connections such as Residual networks [He et al., 2016], GoogLeNet [Szegedy et al., 2015] and DenseNets [Huang et al., 2017] were incepted. Naturally, many of the existing RC models already have some built-in designs to workaround this issue by shortening the signal path in the network. Examples include attention flow

[Seo et al., 2016], residual connections [Xiong et al., 2017; Yu et al., 2018] or simply the usage of highway encoders [Srivastava et al., 2015]. As such, we hypothesize that explicitly improving information flow can lead to further and considerable improvements in RC models.

A second observation is that the flow of $P/Q$ representations across the network are often well-aligned and 'synchronous', i.e., $P$ is often only matched with $Q$ at the same hierarchical stage (e.g., only after they have passed through a fixed number of encoder layers). To this end, we hypothesize that increasing the number of interaction interfaces, i.e., matching in an asynchronous, cross-hierarchical fashion, can also lead to an improvement in performance.

Based on the above mentioned intuitions, this paper proposes a new architecture with two distinct characteristics. Firstly, our network is densely connected, connecting every layer of $P$ with every layer of $Q$. This not only facilitates information flow but also increases the number of interaction interfaces between $P/Q$. Secondly, our network is densely connected by *attention*, making it vastly different from any residual mitigation strategy in the literature. To the best of our knowledge, this is the first work that explicitly considers attention as a form of skip-connector.

Notably, models such as BiDAF incorporates a form of attention propagation (flow). However, this is inherently unsuitable for forging dense connections throughout the network since this would incur a massive increase in the representation size in subsequent layers. To this end, we propose efficient *Bidirectional Attention Connectors* (BAC) as a base building block to connect two sequences at arbitrary layers. The key idea is to compress the attention outputs so that they can be small enough to propagate, yet enabling a connection between two sequences. The propagated features are collectively passed into prediction layers, which effectively connect shallow layers to deeper layers. Therefore, this enables multiple bidirectional attention calls to be executed without much concern, allowing us to efficiently connect multiple layers together.

Overall, we propose DECAPROP (Densely Connected Attention Propagation), a novel architecture for reading comprehension. DECAPROP achieves a significant gain of $2.6\% - 14.2\%$ absolute improvement in F1 score over the existing state-of-the-art on four challenging RC datasets, namely NewsQA [Trischler et al., 2016], Quasar-T [Dhingra et al., 2017], SearchQA [Dunn et al., 2017] and NarrativeQA [Kočiský et al., 2017].

## 2   Bidirectional Attention Connectors (BAC)

This section introduces the Bidirectional Attention Connectors (BAC) module which is central to our overall architecture. The BAC module can be thought of as a *connector* component that connects two sequences/layers.

The key goals of this module are to (1) connect any two layers of $P/Q$ in the network, returning a residual feature that can be propagated[2] to deeper layers, (2) model cross-hierarchical interactions between $P/Q$ and (3) minimize any costs incurred to other network components such that this component may be executed multiple times across all layers.

Let $P \in \mathbb{R}^{\ell_p \times d}$ and $Q \in \mathbb{R}^{\ell_q \times d}$ be inputs to the BAC module. The initial steps in this module remains identical to standard bi-attention in which an affinity matrix is constructed between $P/Q$. In our bi-attention module, the affinity matrix is computed via:

$$E_{ij} = \frac{1}{\sqrt{d}} \mathbf{F}(p_i)^\top \mathbf{F}(q_j) \tag{1}$$

where $\mathbf{F}(.)$ is a standard dense layer with ReLU activations and $d$ is the dimensionality of the vectors. Note that this is the scaled dot-product attention from Vaswani et al. [2017]. Next, we learn an alignment between $P/Q$ as follows:

$$A = \text{Softmax}(E^\top)P \quad \text{and} \quad B = \text{Softmax}(E)Q \tag{2}$$

where $A, B$ are the aligned representations of the query/passsage respectively. In many standard neural QA models, it is common to pass an augmented[3] matching vector of this attentional representation to

subsequent layers. For this purpose, functions such as $f = W([b_i ; p_i; b_i \odot p_i, b_i - p_i]) + b$ have been used [Wang and Jiang, 2016]. However, simple/naive augmentation would not suffice in our use case. Even without augmentation, every call of bi-attention returns a new $d$ dimensional vector for each element in the sequence. If the network has $l$ layers, then connecting[4] all pairwise layers would require $l^2$ connectors and therefore an output dimension of $l^2 \times d$. This is not only computationally undesirable but also require a large network at the end to reduce this vector. With augmentation, this problem is aggravated. Hence, standard birectional attention is not suitable here.

To overcome this limitation, we utilize a parameterized function $G(.)$ to compress the bi-attention vectors down to scalar.

$$g_i^p = [G([b_i; p_i]); G(b_i - p_i); G(b_i \odot p_i)] \tag{3}$$

where $g_i^p \in \mathbb{R}^3$ is the output (for each element in $P$) of the BAC module. This is done in an identical fashion for $a_i$ and $q_i$ to form $g_i^q$ for each element in Q. Intuitively $g_i^*$ where $* = \{p, q\}$ are the learned scalar attention that is propagated to upper layers. Since there are only three scalars, they will not cause any problems even when executed for multiple times. As such, the connection remains relatively lightweight. This compression layer can be considered as a defining trait of the BAC, differentiating it from standard bi-attention.

Naturally, there are many potential candidates for the function $G(.)$. One natural choice is the standard dense layer (or multiple dense layers). However, dense layers are limited as they do not compute dyadic pairwise interactions between features which inhibit its expressiveness. On the other hand, factorization-based models are known to not only be expressive and efficient, but also able to model low-rank structure well.

To this end, we adopt factorization machines (FM) [Rendle, 2010] as $G(.)$. The FM layer is defined as:

$$G(x) = w_0 + \sum_{i=1}^{n} w_i \, x_i + \sum_{i=1}^{n} \sum_{j=i+1}^{n} \langle v_i, v_j \rangle \, x_i \, x_j \tag{4}$$

where $v \in \mathbb{R}^{d \times k}$, $w_0 \in \mathbb{R}$ and $w_i \in \mathbb{R}^d$. The output $G(x)$ is a scalar. Intuitively, this layer tries to learn pairwise interactions between every $x_i$ and $x_j$ using factorized (vector) parameters $v$. In the context of our BAC module, the FM layer is trying to learn a low-rank structure from the 'match' vector (e.g., $b_i - p_i, b_i \odot p_i$ or $[b_i; p_i]$). Finally, we note that the BAC module takes inspiration from the main body of our CAFE model [Tay et al., 2017] for entailment classification. However, this work demonstrates the usage and potential of the BAC as a residual connector.

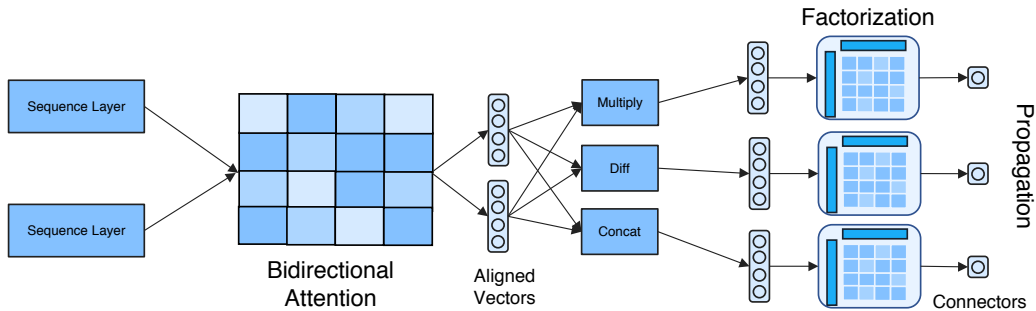

Figure 1: High level overview of our proposed Bidirectional Attention Connectors (BAC). BAC supports connecting two sequence layers with attention and produces connectors that can be propagated to deeper layers of the network.

## 3 Densely Connected Attention Propagation (DECAPROP)

In this section, we describe our proposed model in detail. Figure 2 depicts a high-level overview of our proposed architecture.

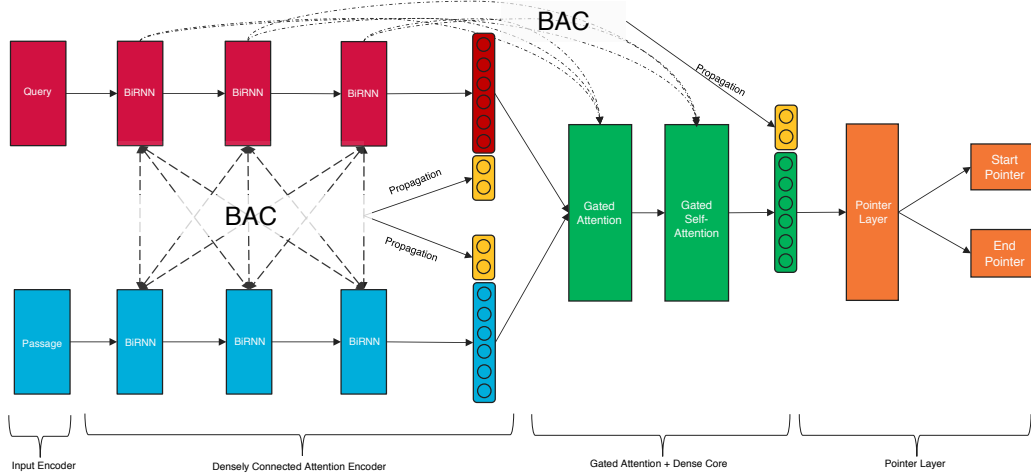

Figure 2: Overview of our proposed model architecture

## 3.1 Contextualized Input Encoder

The inputs to our model are two sequences $P$ and $Q$ which represent passage and query respectively. Given $Q$, the task of the RC model is to select a sequence of tokens in $P$ as the answer. Following many RC models, we enhance the input representations with (1) character embeddings (passed into a BiRNN encoder), (2) a binary match feature which denotes if a word in the query appears in the passage (and vice versa) and (3) a normalized frequency score denoting how many times a word appears in the passage. The Char BiRNN of $h_c$ dimensions, along with two other binary features, is concatenated with the word embeddings $w_i \in \mathbb{R}^{d_w}$, to form the final representation of $d_w + h_c + 2$ dimensions.

## 3.2 Densely Connected Attention Encoder (DECAENC)

The DECAENC accepts the inputs $P$ and $Q$ from the input encoder. DECAENC is a multi-layered encoder with $k$ layers. For each layer, we pass $P/Q$ into a bidirectional RNN layer of $h$ dimensions. Next, we apply our attention connector (BAC) to $H^P/H^Q \in \mathbb{R}^{\approx}$ where $H$ represents the hidden state outputs from the BiRNN encoder where the RNN cell can either be a GRU or LSTM encoder. Let $d$ be the input dimensions of $P$ and $Q$, then this encoder goes through a process of $d \rightarrow h \rightarrow h+3 \rightarrow h$ in which the BiRNN at layer $l + 1$ consumes the propagated features from layer $l$.

Intuitively, this layer models $P/Q$ whenever they are at the same network hierarchical level. At this point, we include 'asynchronous' (cross hierarchy) connections between $P$ and $Q$. Let $P^i, Q^i$ denote the representations of $P, Q$ at layer $i$. We apply the Bidirectional Attention Connectors (BAC) as follows:

$$Z_p^{ij}, Z_q^{ij} = F_C(P^i, Q^j) \quad \forall\, i, j = 1, 2 \cdots n \tag{5}$$

where $F_C$ represents the BAC component. This densely connects all representations of $P$ and $Q$ across multiple layers. $Z_*^{ij} \in \mathbb{R}^{3 \times \ell}$ represents the generated features for each $ij$ combination of $P/Q$. In total, we obtain $3n^2$ compressed attention features for each word. Intuitively, these features capture fine-grained relationships between $P/Q$ at different stages of the network flow. The output of the encoder is the concatenation of all the BiRNN hidden states $H^1, H^2 \cdots H^k$ and $Z^*$ which is a matrix of $(nh + 3n^2) \times \ell$ dimensions.

## 3.3 Densely Connected Core Architecture (DECACORE)

This section introduces the core architecture of our proposed model. This component corresponds to the interaction segment of standard RC model architecture.

**Gated Attention**  The outputs of the densely connected encoder are then passed into a standard gated attention layer. This corresponds to the *'interact'* component in many other popular RC models

that models $Q/P$ interactions with attention. While there are typically many choices of implementing this layer, we adopt the standard gated bi-attention layer following [Wang et al., 2017c].

$$S = \frac{1}{\sqrt{d}} \mathbf{F}(P)^\top (\mathbf{F}(Q)) \tag{6}$$

$$\bar{P} = \text{Softmax}(S)Q \tag{7}$$

$$P' = \text{BiRNN}(\sigma(\mathbf{W}_g([P; \bar{P}]) + \mathbf{b}_g) \odot P) \tag{8}$$

where $\sigma$ is the sigmoid function and $F(.)$ are dense layers with ReLU activations. The output $P'$ is the query-dependent passage representation.

**Gated Self-Attention**   Next, we employ a self-attention layer, applying Equation (8) yet again on $P'$, matching $P'$ against itself to form $B$, the output representation of the core layer. The key idea is that self-attention models each word in the query-dependent passsage representation against all other words, enabling each word to benefit from a wider global view of the context.

**Dense Core**   At this point, we note that there are two intermediate representations of $P$, i.e., one after the gated bi-attention layer and one after the gated self-attention layer. We denote them as $U^1, U^2$ respectively. Unlike the Densely Connected Attention Encoder, we no longer have two representations at each hierarchical level since they have already been 'fused'. Hence, we apply a one-sided BAC to all permutations of $[U^1, U^2]$ and $Q^i$, $\forall i = 1, 2 \cdots k$. Note that the one-sided BAC only outputs values for the left sequence, ignoring the right sequence.

$$R^{kj} = F'_C(U^j, Q^k) \quad \forall k = 1, 2 \cdots n, \forall j = 1, 2 \tag{9}$$

where $R^{kj} \in \mathbb{R}^{3 \times \ell}$ represents the connection output and $F'_C$ is the one-sided BAC function. All values of $R^{kj}$, $\forall j = 1, 2$, $\forall k = 1, 2 \cdots n$ are concatenated to form a matrix $R'$ of $(2n \times 6) \times \ell$, which is then concatenated with $U^2$ to form $M \in \mathbb{R}^{\ell_p \times (d + 12n)}$. This final representation is then passed to the answer prediction layer.

### 3.4   Answer Pointer and Prediction Layer

Next, we pass $M$ through a stacked BiRNN model with two layers and obtain two representations, $H_1^\dagger$ and $H_2^\dagger$ respectively.

$$H_1^\dagger = \text{BiRNN}(M) \text{ and } H_2^\dagger = \text{BiRNN}(H_1^\dagger) \tag{10}$$

The start and end pointers are then learned via:

$$p^1 = \text{Softmax}(w_1 H_1^\dagger) \text{ and } p^2 = \text{Softmax}(w_2 H_2^\dagger) \tag{11}$$

where $w_1, w_2 \in \mathbb{R}^d$ are parameters of this layer. To train the model, following prior work, we minimize the sum of negative log probabilities of the start and end indices:

$$L(\theta) = -\frac{1}{N} \sum_i^N \log(p_{y_i^1}^1) + \log(p_{y_i^2}^2) \tag{12}$$

where $N$ is the number of samples, $y_i^1, y_i^2$ are the true start and end indices. $p_k$ is the k-th value of the vector $p$. The test span is chosen by finding the maximum value of $p_k^1, p_l^2$ where $k \leq l$.

## 4   Experiments

This section describes our experiment setup and empirical results.

### 4.1   Datasets and Competitor Baselines

We conduct experiments on four challenging QA datasets which are described as follows:

**NewsQA** This challenging RC dataset [Trischler et al., 2016] comprises $100k$ QA pairs. Passages are relatively long at about $600$ words on average. This dataset has also been extensively used in benchmarking RC models. On this dataset, the key competitors are BiDAF [Seo et al., 2016], Match-LSTM [Wang and Jiang, 2016], FastQA/FastQA-Ext [Weissenborn et al., 2017], R2-BiLSTM [Weissenborn, 2017], AMANDA [Kundu and Ng, 2018].

**Quasar-T** This dataset [Dhingra et al., 2017] comprises $43k$ factoid-based QA pairs and is constructed using ClueWeb09 as its backbone corpus. The key competitors on this dataset are BiDAF and the Reinforced Ranker-Reader ($R^3$) [Wang et al., 2017a]. Several variations of the ranker-reader model (e.g., SR, $SR^2$), which use the Match-LSTM underneath, are also compared against.

**SearchQA** This dataset [Dunn et al., 2017] aims to emulate the search and retrieval process in question answering applications. The challenge involves reasoning over multiple documents. In this dataset, we concatenate all documents into a single passage context and perform RC over the documents. The competitor baselines on this dataset are Attention Sum Reader (ASR) [Kadlec et al., 2016], Focused Hierarchical RNNs (FH-RNN) [Ke et al., 2018], AMANDA [Kundu and Ng, 2018], BiDAF, AQA [Buck et al., 2017] and the Reinforced Ranker-Reader ($R^3$) [Wang et al., 2017a].

**NarrativeQA** [Kočiský et al., 2017] is a recent QA dataset that involves comprehension over stories. We use the summaries setting[5] which is closer to a standard QA or reading comprehension setting. We compare with the baselines in the original paper, namely Seq2Seq, Attention Sum Reader and BiDAF. We also compare with the recent BiAttention + MRU model [Tay et al., 2018b].

As compared to the popular SQuAD dataset [Rajpurkar et al., 2016], these datasets are either (1) more challenging[6], involves more multi-sentence reasoning or (2) is concerned with searching across multiple documents in an 'open domain' setting (SearchQA/Quasar-T). Hence, these datasets accurately reflect real world applications to a greater extent. However, we regard the concatenated documents as a single context for performing reading comprehension. The evaluation metrics are the EM (exact match) and F1 score. Note that for all datasets, we compare all models solely on the RC task. Therefore, for fair comparison we do not compare with algorithms that use a second-pass answer re-ranker [Wang et al., 2017b]. Finally, to ensure that our model is not a failing case of SQuAD, and as requested by reviewers, we also include development set scores of our model on SQuAD.

## 4.2 Experimental Setup

Our model is implemented in Tensorflow [Abadi et al., 2015]. The sequence lengths are capped at $800/700/1500/1100$ for NewsQA, SearchQA, Quasar-T and NarrativeQA respectively. We use Adadelta [Zeiler, 2012] with $\alpha = 0.5$ for NewsQA, Adam [Kingma and Ba, 2014] with $\alpha = 0.001$ for SearchQA, Quasar-T and NarrativeQA. The choice of the RNN encoder is tuned between GRU and LSTM cells and the hidden size is tuned amongst $\{32, 50, 64, 75\}$. We use the CUDNN implementation of the RNN encoder. Batch size is tuned amongst $\{16, 32, 64\}$. Dropout rate is tuned amongst $\{0.1, 0.2, 0.3\}$ and applied to all RNN and fully-connected layers. We apply variational dropout [Gal and Ghahramani, 2016] in-between RNN layers. We initialize the word embeddings with $300D$ GloVe embeddings [Pennington et al., 2014] and are fixed during training. The size of the character embeddings is set to $8$ and the character RNN is set to the same as the word-level RNN encoders. The maximum characters per word is set to $16$. The number of layers in DECAENC is set to $3$ and the number of factors in the factorization kernel is set to $64$. We use a learning rate decay factor of $2$ and patience of $3$ epochs whenever the EM (or ROUGE-L) score on the development set does not increase.

## 5 Results

Overall, our results are optimistic and promising, with results indicating that DECAPROP achieves state-of-the-art performance[7] on all four datasets.

| | Dev | | Test | |
|---|---|---|---|---|
| Model | EM | F1 | EM | F1 |
| Match-LSTM | 34.4 | 49.6 | 34.9 | 50.0 |
| BARB | 36.1 | 49.6 | 34.1 | 48.2 |
| BiDAF | N/A | N/A | 37.1 | 52.3 |
| Neural BoW | 25.8 | 37.6 | 24.1 | 36.6 |
| FastQA | 43.7 | 56.4 | 41.9 | 55.7 |
| FastQAExt | 43.7 | 56.1 | 42.8 | 56.1 |
| R2-BiLSTM | N/A | N/A | 43.7 | 56.7 |
| AMANDA | 48.4 | 63.3 | 48.4 | 63.7 |
| DECAPROP | **52.5** | **65.7** | **53.1** | **66.3** |

Table 1: Performance comparison on NewsQA dataset.

| | Dev | | Test | |
|---|---|---|---|---|
| | EM | F1 | EM | F1 |
| GA | 25.6 | 25.6 | 26.4 | 26.4 |
| BiDAF | 25.7 | 28.9 | 25.9 | 28.5 |
| SR | N/A | N/A | 31.5 | 38.5 |
| $SR^2$ | N/A | N/A | 31.9 | 38.8 |
| $R^3$ | N/A | N/A | 34.2 | 40.9 |
| DECAPROP | **39.7** | **48.1** | **38.6** | **46.9** |

Table 2: Performance comparison on Quasar-T dataset.

| | Dev | | Test | |
|---|---|---|---|---|
| | Acc | $F1^n$ | Acc | $F1^n$ |
| TF-IDF max | 13.0 | N/A | 12.7 | N/A |
| ASR | 43.9 | 24.2 | 41.3 | 22.8 |
| FH-RNN | 49.6 | 56.7 | 46.8 | 53.4 |
| AMANDA | 48.6 | 57.7 | 46.8 | 56.6 |
| DECAPROP | **64.5** | **71.9** | **62.2** | **70.8** |

Table 3: Evaluation on *original setting*, Unigram Accuracy and N-gram F1 scores on SearchQA dataset.

| | Dev | | Test | |
|---|---|---|---|---|
| | EM | F1 | EM | F1 |
| BiDAF | 31.7 | 37.9 | 28.6 | 34.6 |
| AQA | 40.5 | 47.4 | 38.7 | 45.6 |
| $R^3$ | N/A | N/A | 49.0 | 55.3 |
| DECAPROP | **58.8** | **65.5** | **56.8** | **63.6** |

Table 4: Evaluation on Exact Match and F1 Metrics on SearchQA dataset.

| | Test / Validation | | | |
|---|---|---|---|---|
| | BLEU-1 | BLEU-4 | METEOR | ROUGE-L |
| Seq2Seq | 15.89 / 16.10 | 1.26 / 1.40 | 4.08 / 4.22 | 13.15 / 13.29 |
| Attention Sum Reader | 23.20 / 23.54 | 6.39 / 5.90 | 7.77 / 8.02 | 22.26 / 23.28 |
| BiDAF | 33.72 / 33.45 | 15.53 / 15.69 | 15.38 / 15.68 | 36.30 / 36.74 |
| BiAttention + MRU | - / 36.55 | - /19.79 | - / 17.87 | - / 41.44 |
| DECAPROP | **42.00 / 44.35** | **23.42 / 27.61** | **23.42 / 21.80** | **40.07 / 44.69** |

Table 5: Evaluation on NarrativeQA (Story Summaries).

| Model | EM | F1 |
|---|---|---|
| DCN [Xiong et al., 2016] | 66.2 | 75.9 |
| DCN + CoVE [McCann et al., 2017] | 71.3 | 79.9 |
| R-NET (Wang et al.) [Wang et al., 2017c] | 72.3 | 80.6 |
| R-NET (Our re-implementation) | 71.9 | 79.6 |
| DECAPROP (This paper) | 72.9 | 81.4 |
| QANet [Yu et al., 2018] | 73.6 | 82.7 |

Table 6: Single model dev scores (published scores) of some representative models on SQuAD.

**NewsQA** Table 1 reports the results on NewsQA. On this dataset, DECAPROP outperforms the existing state-of-the-art, i.e., the recent AMANDA model by ($+4.7\%$ EM / $+2.6\%$ F1). Notably, AMANDA is a strong neural baseline that also incorporates gated self-attention layers, along with question-aware pointer layers. Moreover, our proposed model also outperforms well-established baselines such as Match-LSTM ($+18\%$ EM / $+16.3\%$ F1) and BiDAF ($+16\%$ EM / $+14\%$ F1).

**Quasar-T** Table 2 reports the results on Quasar-T. Our model achieves state-of-the-art performance on this dataset, outperforming the state-of-the-art $R^3$ (Reinforced Ranker Reader) by a considerable margin of $+4.4\%$ EM / $+6\%$ F1. Performance gain over standard baselines such as BiDAF and GA are even larger ($> 15\%$ F1).

**SearchQA** Table 3 and Table 4 report the results[8] on SearchQA. On the original setting, our model outperforms AMANDA by $+15.4\%$ EM and $+14.2\%$ in terms of F1 score. On the overall setting, our model outperforms both AQA ($+18.1\%$ EM / $+18\%$ F1) and Reinforced Reader Ranker ($+7.8\%$ EM /

+8.3% F1). Both models are reinforcement learning based extensions of existing strong baselines such as BiDAF and Match-LSTM.

**NarrativeQA**    Table 5 reports the results on NarrativeQA. Our proposed model outperforms all baseline systems (Seq2Seq, ASR, BiDAF) in the original paper. On average, there is a $\approx +5\%$ improvement across all metrics.

**SQuAD**    Table 6 reports dev scores[9] of our model against several representative models on the popular SQuAD benchmark. While our model does not achieve state-of-the-art performance, our model can outperform the base R-NET (both our implementation as well as the published score). Our model achieves reasonably competitive performance.

## 5.1    Ablation Study

We conduct an ablation study on the NewsQA development set (Table 7). More specifically, we report the development scores of seven ablation baselines. In (1), we removed the entire DECAPROP architecture, reverting it to an enhanced version of the original R-NET model[10]. In (2), we removed DECACORE and passed $U^2$ to the answer layer instead of $M$. In (3), we removed the DECAENC layer and used a 3-layered BiRNN instead. In (4), we kept the DECAENC but only compared layer of the same hierarchy and omitted cross hierarchical comparisons. In (5), we removed the Gated Bi-Attention and Gated Self-Attention layers. Removing these layers simply allow previous layers to pass through. In (6-7), we varied $n$, the number of layers of DECAENC. Finally, in (8-9), we varied the FM with linear and nonlinear feed-forward layers.

| Ablation | EM | F1 |
|---|---|---|
| (1) Remove All (R-NET) | 48.1 | 61.2 |
| (2) w/o DECACORE | 51.5 | 64.5 |
| (3) w/o DECAENC | 49.3 | 62.0 |
| (4) w/o Cross Hierarchy | 50.0 | 63.1 |
| (5) w/o Gated Attention | 49.4 | 62.8 |
| (6) Set DECAENC $n = 2$ | 50.5 | 63.4 |
| (7) Set DECAENC $n = 4$ | 50.7 | 63.3 |
| (8) DecaProp (Linear d->1) | 50.9 | 63.0 |
| (9) DecaProp (Nonlinear d->d->1) | 48.9 | 60.0 |
| Full Architecture ($n = 3$) | **52.5** | **65.7** |

Table 7: Ablation study on NewsQA development set.

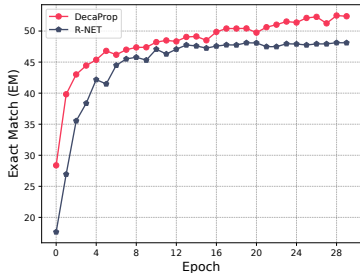

Table 8: Development EM score (DECAPROP versus R-NET) on NewsQA.

From (1), we observe a significant gap in performance between DECAPROP and R-NET. This demonstrates the effectiveness of our proposed architecture. Overall, the key insight is that all model components are crucial to DECAPROP. Notably, the DECAENC seems to contribute the most to the overall performance. Finally, Figure 8 shows the performance plot of the development EM metric (NewsQA) over training. We observe that the superiority of DECAPROP over R-NET is consistent and relatively stable. This is also observed across other datasets but not reported due to the lack of space.

## 6    Related Work

In recent years, there has been an increase in the number of annotated RC datasets such as SQuAD [Rajpurkar et al., 2016], NewsQA [Trischler et al., 2016], TriviaQA [Joshi et al., 2017] and RACE

[Lai et al., 2017]. Spurred on by the avaliability of data, many neural models have also been proposed to tackle these challenges. These models include BiDAF [Seo et al., 2016], Match-LSTM [Wang and Jiang, 2016], DCN/DCN+ [Xiong et al., 2016, 2017], R-NET [Wang et al., 2017c], DrQA [Chen et al., 2017], AoA Reader [Cui et al., 2016], Reinforced Mnemonic Reader [Hu et al., 2017], ReasoNet [Shen et al., 2017], AMANDA [Kundu and Ng, 2018], $R^3$ Reinforced Reader Ranker [Wang et al., 2017a] and QANet [Yu et al., 2018]. Many of these models innovate at either (1) the bidirectional attention layer (BiDAF, DCN), (2) invoking multi-hop reasoning (Mnemonic Reader, ReasoNet), (3) reinforcement learning ($R^3$, DCN+), (4) self-attention (AMANDA, R-NET, QANet) and finally, (5) improvements at the encoder level (QANet). While not specifically targeted at reading comprehension, a multitude of pretraining schemes [McCann et al., 2017; Peters et al., 2018; Radford et al.; Devlin et al., 2018] have recently proven to be very effective for language understanding tasks.

Our work is concerned with densely connected networks aimed at improving information flow [Huang et al., 2017; Srivastava et al., 2015; Szegedy et al., 2015]. While most works are concerned with computer vision tasks or general machine learning, there are several notable works in the NLP domain. Ding et al. [2018] proposed Densely Connected BiLSTMs for standard text classification tasks. [Tay et al., 2018a] proposed a co-stacking residual affinity mechanims that includes all pairwise layers of a text matching model in the affinity matrix calculation. In the RC domain, DCN+ [Xiong et al., 2017] used Residual Co-Attention encoders. QANet [Yu et al., 2018] used residual self-attentive convolution encoders. While the usage of highway/residual networks is not an uncommon sight in NLP, the usage of bidirectional attention as a skip-connector is new. Moreover, our work introduces new cross-hierarchical connections, which help to increase the number of interaction interfaces between $P/Q$.

## 7 Conclusion

We proposed a new *Densely Connected Attention Propagation* (DECAPROP) mechanism. For the first time, we explore the possibilities of using birectional attention as a skip-connector. We proposed Bidirectional Attention Connectors (BAC) for efficient connection of any two arbitrary layers, producing connectors that can be propagated to deeper layers. This enables a shortened signal path, aiding information flow across the network. Additionally, the modularity of the BAC allows it to be easily equipped to other models and even other domains. Our proposed architecture achieves state-of-the-art performance on four challenging QA datasets, outperforming strong and competitive baselines such as Reinforced Reader Ranker ($R^3$), AMANDA, BiDAF and R-NET.

## 8 Acknowledgements

This paper is partially supported by Baidu I$^2$R Research Centre, a joint laboratory between Baidu and A-Star I$^2$R. The authors would like to thank the anonymous reviewers of NeuRIPS 2018 for their valuable time and feedback!

## Footnotes

*Denotes equal contribution

[2]Notably, signals still have to back-propagate through the BAC parameters. However, this still enjoys the benefits when connecting far away layers and also by increasing the number of pathways.

[3]This refers to common element-wise operations such as the subtraction or multiplication.

[4]See encoder component of Figure 2 for more details.

[5]Notably, a new SOTA was set by [Hu et al., 2018] after the NIPS submission deadline.

[6]This is claimed by authors in most of the dataset papers.

[7]As of NIPS 2018 submission deadline.

[8] The original SearchQA paper [Dunn et al., 2017], along with AMANDA [Kundu and Ng, 2018] report results on Unigram Accuracy and N-gram F1. On the other hand, [Buck et al., 2017] reports results on overall EM/F1 metrics. We provide comparisons on both.

[9]Early testing of our model was actually done on SQuAD. However, since taking part on the heavily contested public leaderboard requires more computational resources than we could muster, we decided to focus on other datasets. In lieu of reviewer requests, we include preliminary results of our model on SQuAD dev set.

[10]For fairer comparison, we make several enhancements to the R-NET model as follows: (1) We replaced the additive attention with scaled dot-product attention similar to ours. (2) We added shortcut connections after the encoder layer. (3) We replaced the original Pointer networks with our BiRNN Pointer Layer. We found that these enhancements consistently lead to improved performance. The original R-NET performs at $\approx 2\%$ lower on NewsQA.

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
