[Reviews · NeurIPS 2018]

Reviewer 1



This paper proposes a new neural architecture for reading comprehension. Compared to many other existing neural architectures, this model 1) densely connects all pairs of passage layers and question layers for encoding; 2) uses a component called Bidirectional Attention Connectors (BAC) for connecting any P-layer and Q-layer which employs an FM layer on top of commonly used bi-directional attention. The proposed architecture has been evaluated on four reading comprehension datasets and demonstrates strong empirical results. Overall, although the proposed ideas could be potentially interesting, I think the presented results (in the current presentation format) are not convincing enough. - I am not sure how the four datasets were chosen, but this model is not evaluated on the most competitive datasets: SQuAD and (arguably) TriviaQA. This makes the results less convincing. - In general, I find it is difficult to understand the influence of the proposed components in this paper. According to the ablation study in Table 6, I don’t see any of the components contributed that much to the overall performance and whether the results can generalize to other datasets. I think it is necessary to have more ablation studies on other datasets and apparently, this model has also improved more on other datasets. Gated attention is not novel for this paper. How useful are these dense connections are indeed? How useful is the FM layer (if we just replace function G with a feedforward network)? I think it is important to highlight the new components of this paper and their contributions to a reading comprehension system (on various datasets). - I think the paper could have done a better job explaining the intuitions of the proposed solutions. The paper made a few claims in the introduction part such as “densely connected by attention” and “compress the attention outputs so that they can be small enough to propagate” but it is unclear how they are reflected in the model design. - The notions in this paper are also confusing. - Line 76 and equation (3): how are b_i and p_i defined? - How are Z* used/propagated in the following layers? ========= I have carefully read all other reviews and the authors' rebuttal. Given the authors provided results on SQuAD and comparisons of FM vs feedforward networks, I increased my score from 5 to 6.

Reviewer 2



Given the authors' response with additional data points on SQuAD and ablation tests for FM, I have gladly increased the score to 7. Good luck! --- This paper has presented several novel ideas for reading comprehension (RC) that I really like: (a) the Bidirectional Attention Connectors (BAC), which captures interaction between two sequences from the "attention" point-of-view and then cheaply compresses the output into a few (3) scalars. BAC seems to be broadly useful for many networks. (b) Densely connect layers: an inspiration from DenseNet applied to RC The authors also presented strong results on 4 RC benchmarks and have an ablation test which looks convincing. I also appreciate the authors reported on the hyperparameters they tune over. The reason that I give this submission a score of 6 is because there's no results on SQuAD. I really don't buy the argument that the other 4 datasets are more challenging, so there's no point trying SQuAD. SQuAD has a lot of baselines to be compared over as already cited by the authors, so it's very natural to know how much these innovations bring. I'm confident on my rating because I do research on RC as well. A few other questions for the authors: (a) The choice of factorization machines (FM) seems interesting but also arbitrary. Do the authors have results that replace FM with other choices, e.g., a feedforward network? (b) For the FM formula (lines 97-98), is k=n? (c) For DecaCore, instead of [U1, U2, Q_i], why don't we consider [U1, U2, Q_i, P_i] ? (d) For the normalized frequency feature, is it applied to both the passage and the query?

Reviewer 3



The paper proposes a new span prediction model for reading comprehension (RC) together with Bidirectional Attention Connectors (BAC) which densely connects pairwise layers (outputs of two RNN) in the model. The paper conduct experiments on 4 RC datasets and shows that their model provides state of the art performance on all the tasks. The BAC generalizes matrix attention used often in RC models and outputs a low dimensional representation for each timestep (dim=3) employing factorization machines (from previous work) for the projection to lower dimension, which explicitly models the interaction of features in the input vector. The paper presentation is clear (although math needs a lot of proofreading, see below), it presents the model, comments on the choice of the datasets, experimental setup, and also briefly describes the SOTA models for the RC tasks attempted. Moreover, the ablation study of features of the model suggests that all parts are important. - The hidden layer sizes tried in tuning (up to 75 dim) seem a bit small, but the multi layer architecture seems to make up for that. Would be interesting to see less layers with larger hidden sizes (e.g. 128, 256) for comparison. - It would be nice if the ablation included using a simple linear projection instead of the factorization machines approach. - Eq. 2: the transpose might be incorrect since E is l_p x l_q, please double check this. - Line 117, H^P, H^Q in R (this is probably a vector) - Eq 5 quantifies over i,j but there is no i in the expression - Line 126: should Z^*_{kj} be Z^{kj}_* ? (* for p,q) to match Eq 5 notation. - Section 3.2 uses two types of the letter ‘l’, and l_p and l_q is used in previous sections for similar quantities, which is confusing. - Eq 6, extra parenthesis - Eq 12 has p^1 and p^2, should these be p^s, p^e as in Eq 11?